# The Role of Plasma Cells as a Marker of Chronic Endometritis: A Systematic Review and Meta-Analysis

**DOI:** 10.3390/biomedicines11061714

**Published:** 2023-06-15

**Authors:** Angela Santoro, Antonio Travaglino, Frediano Inzani, Giuseppe Angelico, Antonio Raffone, Giuseppe Maria Maruotti, Patrizia Straccia, Damiano Arciuolo, Federica Castri, Nicoletta D’Alessandris, Giulia Scaglione, Michele Valente, Federica Cianfrini, Valeria Masciullo, Gian Franco Zannoni

**Affiliations:** 1Unità di Ginecopatologia e Patologia Mammaria, Dipartimento Scienze della Salute della Donna, del Bambino e di Sanità Pubblica, Fondazione Policlinico Universitario A. Gemelli-IRCCS, Largo A. Gemelli 8, 00168 Rome, Italy; 2Anatomic Pathology Unit, Department of Molecular Medicine, University of Pavia and Fondazione IRCCS San Matteo Hospital, 27100 Pavia, Italy; 3Department of Medical and Surgical Sciences and Advanced Technologies “G. F. Ingrassia”, Anatomic Pathology, University of Catania, 95123 Catania, Italy; 4Gynecology and Obstetrics Unit, Department of Public Health, University of Naples Federico II, 80138 Naples, Italy; 5Division of Gynecologic Surgery, Department of Woman, Child and Public Health, Fondazione Policlinico Universitario A. Gemelli-IRCCS, 00168 Rome, Italy; 6Istituto di Anatomia Patologica, Università Cattolica del Sacro Cuore, Largo A. Gemelli 8, 00168 Rome, Italy

**Keywords:** chronic endometritis, infertility, hysteroscopy, endometrial biopsy, pregnancy

## Abstract

Chronic endometritis (CE) is the persistent inflammation of the endometrial lining associated with infertility and various forms of reproductive failures. The diagnosis of CE is based on the histological evidence of stromal plasma cells; however, standardized methods to assess plasma cells are still lacking. In the present paper, we aimed to determine the most appropriate plasma cell threshold to diagnose CE based on pregnancy outcomes. Three electronic databases were searched from their inception to February 2022 for all studies comparing pregnancy outcomes between patients with CE and patients without CE. The relative risk (RR) of pregnancy, miscarriage, and/or live birth rates were calculated and pooled based on the plasma cell threshold adopted. A *p*-value < 0.05 was considered significant. Nine studies adopting different thresholds (1 to 50 plasma cells/10 HPF) were included. In the meta-analysis, we only found a significant association between miscarriage rate and a plasma cell count ≥ 5/10 HPF (RR = 2.4; *p* = 0.007). Among studies not suitable for meta-analysis, CE showed an association with worsened pregnancy only when high thresholds (10 and 50/10 HPF) were adopted. In conclusion, our study suggests that the presence of plasma cells at low levels (<5/10 HPF) may not predict worsened pregnancy outcomes. Based on these findings, a threshold of ≥5 plasma cells/10 HPF may be more appropriate to diagnose CE.

## 1. Introduction

Infertility is a common gynecological disease whose incidence has been constantly increasing in recent years [1]. The most relevant factors for predicting the success of embryo implantation and pregnancy rates include embryo quality and endometrial receptivity [2].

Inflammatory conditions affecting the endometrial cavity may influence fertilized egg implantation, resulting in infertility or miscarriage [3,4].

Physiologically, the endometrium contains a large variety of immunocompetent stromal cells, including natural killers (NKs), macrophages, T cells, and neutrophils, and its composition and density fluctuate periodically during the menstrual cycle [5,6]. Whereas leucocytes account for less than 10% of the stromal cells in the proliferative and early secretory phases, their numbers increase dramatically from the mid-secretory phase, increasing further in the late secretory phase and early pregnancy [5,6]. These subpopulations’ cycle-dependent changes play a crucial role in implantation. Previous work suggested that maternal leukocyte populations in the endometria of some women with recurrent miscarriages are significantly different from those carrying the fetuses to term or associated with known fertility [5,6]. Among the immune cell populations present in the endometrial stroma, high uterine natural killer (uNK) cell and plasma cell counts have been reported to be associated with recurrent miscarriage (RM) and recurrent implantation failure (RIF) or repeated in vitro fertilization (IVF) failure [7,8]. Generally, the possible underlying causes of RM and RIF are represented by uterine anatomical abnormalities, parental karyotype alterations, clotting disorders such as protein C deficiency, factor V Leiden mutation, and antiphospholipid syndrome [9,10]. However, in approximately 50% of women with RM and RIF, the problem remains “unluckily unexplained” and is probably related to chronic endometritis (CE) [4,11].

CE is a persistent inflammation of the endometrial lining, whose estimated incidence in infertile women ranges from 0.2 to 46% [4,11,12,13].

Histologically, CE is characterized by the presence of inflammatory cells in the endometrial stroma, including plasma cells, lymphocytes, eosinophils, and even lymphoid follicles [4,11,12,13]. Other histopathologic characteristics of CE include superficial edema, increased stromal density, and unsynchronized differentiation between endometrial epithelium and stroma. The most common putative organisms isolated in endometrial samples from patients affected by CE are Streptococcus, Enterococcus, Escherichia, and Staphylococcus as most prevalent but also *Chlamydia trachomatis* (Chlamydia), *Neisseria gonorrhoeae* (Gonorrhea), *Mycoplasma hominis*, *Mycobacterium tuberculosis*, and several viruses [4,11,12,13]. 

The most effective treatment is represented by antibiotic therapy, which is curative in the majority of cases. However, untreated patients may develop chronic pelvic inflammatory disease (PID) [4,11,12,13]. Severe forms of CE also include chronic granulomatous endometritis caused by tuberculosis and pyometra, which is a form of CE affecting elderly women associated with stenosis of the cervical os and the accumulation of discharges and infection [4,11,12,13]. Despite being related to infertility and the increased risk of spontaneous abortion, CE is usually asymptomatic and only rarely accompanied by aspecific symptoms, including pelvic pain, dysfunctional uterine bleeding, dyspareunia, and leucorrhea [4,11,12,13]. 

Although there has been increased scientific interest in the putative relationship between CE, infertility, a reduction in uterine receptivity, and the negative impact on reproductive outcomes (recurrent miscarriage RM, recurrent implantation failure RIF, low pregnancy or birth rate), strict clinical and histopathological diagnostic criteria have not yet been fully clarified.

The gold standard for the diagnosis of CE is represented by hysteroscopy with endometrial biopsy [14]. Histologically, CE can be confirmed based on finding a plasma cell infiltrate within the endometrium [14,15]. However, to date, there is a conspicuous lack of standardized methods for the histological assessment of plasma cell infiltrate, and several thresholds for plasma cell count have been proposed in the literature [4,11,12,13,14,15]. In this regard, according to some authors, the histological evidence of one or more plasma cells within an endometrial biopsy is sufficient for the diagnosis of CE; however, other authors have adopted different thresholds arguing that multiple plasma cells should be observed in order to diagnose CE. Moreover, despite immunohistochemistry for CD138 being more sensitive in the identification of plasma cells, several studies still based their findings on the identification of plasma cells with hematoxylin and eosin (H & E) staining [4,11,12,13].

Given the lack of standardized methods for the histological assessment of plasma cell infiltrate and the still limited knowledge on the role of CE in infertility, in the present paper, we aimed to collect the most relevant literature data regarding the diagnosis of CE in infertile women with a particular focus on the most clinically relevant histological threshold for plasma cell count. Moreover, we aimed to observe which plasma cell threshold was significantly associated with miscarriage.

## 2. Materials and Methods

Methods for this study were defined a priori based on previous studies [16,17,18]. Each step of the systematic review and meta-analysis was performed by two independent authors, who consulted each other at the end of each step. Data analysis was performed by using Statistical Package for Social Science (SPSS) 18.0 package (SPSS Inc., Chicago, IL, USA). This study was reported according to the PRISMA statement [19].

### 2.1. Electronic Search and Study Selection

Three electronic databases (Scopus, PubMed, and Web of Science) were consulted from the inception of each database to February 2022. The keywords used were endometritis, plasma cells, and CD138. 

The literature search strategy, as well as article selection, is outlined in Figure 1, a flow chart of the systematic literature search according to PRISMA guidelines.

### 2.2. Study Selection and Data Extraction

Three authors (A.S., F.I., and A.T.) independently performed the study selection. Disagreements were discussed with a third reviewer (G.A.). Data extraction was performed independently by two authors (A.T. and A.S.). A manual search of the reference list of each study was performed to avoid missing relevant publications. One author (G.A.) reviewed the selection and data extraction process. Results were compared, and any disagreement was resolved by consensus.

Data were extracted according to PICO [19]: P (population)—premenopausal women assessed for CE; I (intervention or risk factor)—CE diagnosed at the pathological examination; C (comparator)—NCE at the pathological examination; and O (outcome)—pregnancy, miscarriage and/or live birth rates. Secondary data extracted were country, selection criteria, the period of enrollment, sample size, the percentage of CE, and criteria for defining CE. Studies were grouped according to the plasma cell threshold used to define CE.

### 2.3. Inclusion and Exclusion Criteria

In the present meta-analysis, we adopted the following inclusion criteria: (i) studies comparing pregnancy, miscarriage, and/or live birth rates between patients with CE and patients with NCE and (ii) studies where the diagnosis of CE was achieved based on histological plasma cell count. 

On the other hand, exclusion criteria were (i) studies regarding patients with a clinical diagnosis of CE not histologically confirmed by plasma cell count; (ii) studies with overlapping patient data; (iii) reviews; and (iv) studies evaluating other types of endometrial inflammation including acute, subacute, and tubercular endometritis.

Reference lists from relevant studies were also searched.

All original studies (experimental and observational) reported in the English language were evaluated.

### 2.4. Risk of Bias Assessment

The risk of bias within studies was assessed using QUADAS-2 [20]. Four domains were assessed: (1) patient selection (were selection criteria and period of enrollment exhaustively reported?); (2) index test (were immunohistochemical criteria exhaustively reported?); (3) reference standard (were pregnancy outcomes exhaustively assessed and reported?); and (4) flow and timing (were patients adequately followed to assess pregnancy outcomes?). The risk of bias was categorized as “low”, “high”, or “unclear” in each domain.

### 2.5. Data Analysis

Relative risk (RR) with a 95% confidence interval (CI) was used to assess the association of CE with pregnancy, miscarriage, and/or live birth rates. When two or more studies adopted the same plasma cell threshold and were comparable, RRs were pooled using a random-effect model. Results were graphically reported on forest plots. Statistical heterogeneity among studies was quantified using the inconsistency index I^2^, as previously described [16,17,18]. A *p*-value < 0.05 was considered significant. Review Manager 5.3 (Copenhagen: The Nordic Cochrane Centre, Cochrane Collaboration, 2014) was used for the analysis.

## 3. Results

### 3.1. Study Selection and Characteristics

Nine studies were included in the systematic review at the end of the study selection process (Figure 1) [1,2,13,15,21,22,23,24,25]. Selection criteria varied among the included studies (Table 1). Regarding the criteria for defining CE, the plasma cell threshold adopted were 5/HPF (equating to 50/10 HPF) in one study, 1/HPF (equating to 10/10 HPF) in four studies, 0.25/HPF (equating to 2.5/10 HPF) in one study, and 1/10 HPF in one study; the remaining two studies assessed several different thresholds (Table 1). Regarding pregnancy outcomes, five studies prospectively assessed pregnancy, miscarriage, and live birth rates. One study assessed pregnancy, miscarriage, and ongoing pregnancy rates. One study prospectively assessed pregnancy rate and retrospectively assessed miscarriages and live births. Two studies performed a retrospective assessment of miscarriages (Table 1).

### 3.2. Risk of Bias Assessment

All studies were considered to have low risks of bias for the patient selection and index test domains (selection criteria, the period of enrollment, and immunohistochemical criteria exhaustively reported). Three studies were considered to have unclear risks of bias for the reference standard domain (data related to one or more outcomes were retrieved from anamnesis), and two studies were considered to have unclear risks for the flow and timing domain (patients were not systematically followed to assess outcomes). Results of the risk of bias assessment are reported in Appendix A.

### 3.3. Statistical Analysis

Meta-analysis showed no significant differences in pregnancy rates between CE and NCE for the 1/10 HPF (RR = 0.81, 95% CI 0.32–2.05; *p* = 0.65), 5/10 HPF (RR = 0.94, 95% CI 0.62–1.44), and 10/10 HPF (RR = 0.80, 95% CI 0.43–1.51; *p* = 0.50) thresholds (Figure 2). A significant association between CE and miscarriage was found for the 5/10 HPF threshold (RR = 2.4, 95% CI 1.29–4.75; *p* = 0.007) but not for the 1/10 HPF (RR = 1.73, 95% CI 0.31–9.68; *p* = 0.54) and 10/10 HPF thresholds (RR = 1.04, 95% CI 0.35–3.12; *p* = 0.94) (Figure 3). No significant differences in live birth rates were found for the 1/10 HPF (RR = 0.81, 95% CI 0.53–1.23; *p* = 0.33), 5/10 HPF (RR = 0.54, 95% CI 0.23–1.28), and 10/10 HPF (RR = 0.96, 95% CI 0.68–1.36; *p* = 0.50) thresholds (Figure 4).

Among individual studies not suitable for meta-analysis, pregnancy rate was significantly associated with CE for the 2.5/10 HPF threshold (RR = 1.34, 95% CI 1.02–1.75; *p* = 0.03) and with NCE for the 50/10 HPF (RR = 0.25, 95% CI 0.06–0.97; *p* = 0.05) threshold (Figure 1); miscarriage was significantly associated with NCE for the 1/10 HPF threshold and with CE for the 10/10 HPF threshold (RR = 1.75, 95% CI 1.40–2.18; *p* < 0.001), while no significant associations were found for the 2.5/10 HPF (RR = 0.65, 95% CI 0.30–1.67; *p* = 0.26) and 50/10 HPF (RR = 1.93, 95% CI 0.74–5.03; *p* = 0.18) thresholds (Figure 2); live birth was significantly associated with CE for the 2.5/10 HPF threshold (RR = 1.39, 95% CI 1.02–1.90; *p* = 0.04) but not for the 50/10 HPF threshold (RR = 0.70, 95% CI 0.39–1.26; *p* = 0.23) (Figure 3).

## 4. Discussion

### 4.1. Main Findings

Our systematic review summarized for the first time the available evidence on the diagnostic threshold for plasma cell count in chronic endometritis. 

The analysis included a total of 2795 infertile patients from 9 observational studies. The overall quality of the included studies was fair since all studies were considered to have low risks of bias for the patient selection and index test domains (selection criteria, the period of enrollment, and immunohistochemical criteria exhaustively reported).

We observed high heterogeneity in the selection criteria and plasma cell threshold used among the several studies. In detail, the plasma cell threshold adopted were 5/HPF (equating to 50/10 HPF) in one study, 1/HPF (equating to 10/10 HPF) in four studies, 0.25/HPF (equating to 2.5/10 HPF) in one study, and 1/10 HPF in one study; the remaining two studies assessed several different thresholds. 

Our meta-analysis showed that the only significant association was found between miscarriage rate and a plasma cell count ≥ 5/10 HPF. Based on individual studies, when CE was defined using low thresholds (i.e., 1/10 HPF and 2.5/10 HPF), it showed no significant results or even association with improved pregnancy outcomes. Significant associations (although inconsistent) were found between CE and worsened pregnancy outcomes when higher thresholds (i.e., 10/10 HPF and 50/10 HPF) were adopted. Moreover, among individual studies not suitable for meta-analysis, pregnancy rate was significantly associated with CE for the 2.5/10 HPF threshold (Figure 1) and with NCE for the 50/10 HPF threshold; miscarriage was significantly associated with NCE for the 1/10 HPF threshold and with CE for the 10/10 HPF threshold, while no significant associations were found for the 2.5/10 HPF and 50/10 HPF thresholds (Figure 2); live birth was significantly associated with CE for the 2.5/10 HPF threshold but not for the 50/10 HPF threshold (Figure 3).

Two recently published studies not included in the meta-analysis adopted two different cut-offs for the diagnosis of CE in patients with RIF:-Five or more CD138^+^ plasma cells in the endometrial stroma detected in more than three HPFs [27].-Five or more CD138^+^ plasma cells in the endometrial stroma in at least one out of thirty randomly selected HPF [28].

In both studies, RIF patients with CE exhibited a statistically significant lower live birth rate compared to RIF patients without CE diagnosis [27,28].

### 4.2. Definition of Acute and Chronic Endometritis

Based on histopathological findings, endometritis can be divided into two subcategories: acute and chronic endometritis. 

Acute endometritis develops as a result of ascending infections from bacteria usually located in the uterine cervix and vagina (*Streptococcus pyogenes*, *Staphylococcus aureus*, *Chlamydia*) [4,11,12,13]. Histopathological findings of acute endometritis include (i) neutrophil micro-abscesses in the endometrium; (ii) neutrophil infiltration in the superficial epithelium; and iii) neutrophil micro-abscesses in the lumen of the glands of the endometrium [4,11,12,13]. The most common clinical findings in acute endometritis include fever, pelvic pain, increased vaginal discharge, abdominal pain and distension, and abnormal vaginal bleeding. The risk of endometritis is higher following uterine surgery, cesarean section, or hysteroscopic procedures if proper asepsis is not maintained. However, endometritis may also develop in patients without risk factors following spontaneous vaginal delivery. 

The most common risk factors for developing acute endometritis include (i) lower socioeconomic status; (ii) high body mass index (BMI); (iii) prolonged rupture of amniotic membranes; (iv) repeated vaginal examinations; (v) chorioamnionitis; and (vi) undiagnosed untreated vaginal infection. 

Diagnostic tests for the diagnosis of acute endometritis include total leucocyte count, a swab culture of the cervix, and a microscopic examination of vaginal discharge samples [4,11,12,13].

CE is a localized inflammatory disorder of the endometrial lining that is often caused by intrauterine infections from microorganisms commonly found in the urogenital area, including *Streptococcus* species, *Staphylococcus* species, *Escherichia coli*, *Enterococcus* faecalis, *Mycobacterium tuberculosis*, *Mycoplasma* species, and *Ureaplasma* species [4,11,12,13]. Antibiotic treatment is effective in eradicating the immunocompetent cells in this pathology, namely the endometrial stromal plasma cells. CE is often associated with female infertility; endometriosis; repeated implantation failure; recurrent pregnancy loss, obstetric complications, such as preeclampsia and preterm labor; and neonatal diseases in premature infants, such as periventricular leukomalacia [4,11,12,13].

CE is often asymptomatic and may be easily overlooked by affected patients and even experienced gynecologists [4,11,12,13]. While acute endometritis manifests with intense symptoms, such as pelvic pain, vaginal discharge, and systemic fever, the clinical course of CE remains largely unknown, including its onset, progress, and remission [4,11,12,13]. 

Etiopathogenesis of CE is still a matter of debate in the scientific community since available literature data are mainly based on observational studies with heterogeneous design and diagnostic criteria; moreover, the relationship between CE, implantation defects, and female infertility has not been fully established. The most accepted theories for explaining the relationship between CE and infertility include (i) altered cytokine and chemokine secretion following activation of local inflammatory processes; (ii) abnormal leukocyte infiltration within the endometrium; (iii) altered uterine contractility; (iv) defective decidualization; and (v) defective endometrial vascularization.

### 4.3. Diagnosis of Chronic Endometritis

CE is traditionally diagnosed by combining endometrial biopsy, histopathology, and immunohistochemistry [4,11,12,13,14,15]. Histopathologic features of CE include increased stromal density, unsynchronized differentiation between endometrial epithelium and stroma, superficial edema, and infiltration of CD138(+) plasma cells, which are the most specific and sensitive findings of CE [4,11,12,13,14,15]. Plasma cells are scattered or clustered cells found in the endometrial stroma. Antibiotic treatment has been shown to effectively eradicate plasma cells, but studies are needed to determine if the histopathologic cure of CE improves reproductive outcomes in affected infertile women [4,11,12,13,14,15].

However, the histological identification of plasma cells may be challenging and time-consuming. The introduction of immunohistochemistry for CD138 significantly improved the sensitivity, specificity, inter-observer variability, and intra-observer variability of CE diagnosis [4,11,12,13,14,15]. Nevertheless, there are some precautions when using immunohistochemistry for CD138. In detail, CD138 can also be expressed in endometrial surface/glandular epithelial cells. Therefore, this positivity in endometrial epithelial cells may result in the overdiagnosis of CE [4,11,12,13,14,15]. Moreover, CD138 immunohistochemistry is not yet standardized for human endometrial tissue. Laboratory factors, such as the dilution and incubation periods of primary antibodies, the section thickness, as well as devices utilized for endometrial biopsy, can influence the diagnosis of CE. 

Histological modifications of the glandular and stromal compartments related to the menstrual cycle, such as mononuclear cell infiltration, increased mitotic activity, and stroma cell proliferation, may also interfere with the identification of plasma cells [14,15].

Moreover, B lymphocytes, which differentiate into plasma cells, are mainly located in the basal layer of the endometrium. Therefore, since the functional layer of the endometrium is thinner during the proliferative phase, there is a higher chance of obtaining more CD138+ plasma cells if endometrial biopsies are taken during the proliferative phase [14,15].

The immunohistochemical expression of MUM1 is emerging as an additional diagnostic tool for the identification of plasma cells in CE. MUM1 (interferon regulatory factor 4) is a transcription factor that is expressed in the late stages of B cell differentiation toward plasmacytes [27]. According to recent studies, MUM1 shows a minor background staining compared to CD138 and a higher inter-observer agreement [29,30]. Therefore, the concomitant use of both IHC-CD138 and IHC-MUM1 may potentially compensate for the shortcomings of each method in the histopathologic diagnosis of chronic endometritis.

### 4.4. Diagnostic Role of Hysteroscopy 

Fluid hysteroscopy has been recently proposed as a versatile and convenient diagnostic tool that can be utilized in gynecologic practice to detect uterine cavity lesions associated with female infertility, such as endometrial polyps, submucosal leiomyomas, intrauterine adhesions, and uterine septum [31]. Recent studies have also demonstrated the potential utility of fluid hysteroscopy for the endoscopic diagnosis of chronic endometritis (CE) since it represents a less painful and time-consuming alternative to traditional endometrial biopsy [31,32].

In 2019, the International Working Group for Standardization of Chronic Endometritis Diagnosis proposed diagnostic criteria for the hysteroscopic assessment of CE [33]. These criteria include five hysteroscopic features: the strawberry aspect of the endometrial lining, focal hyperemia, hemorrhagic spots, micropolyposis, and stromal edema. Among these features, endometrial micropolyposis is the most easily visible and is strongly associated with the presence of histopathologic CE.

In this regard, the retrospective study by Cicinelli et al. showed that endometrial micropolyposis can easily be detected by fluid hysteroscopy; moreover, histopathologic evidence of plasma cells was less frequently identified in women without hysteroscopic evidence of micropolyposis [34]. In another retrospective study, the positive and negative predictive values of fluid hysteroscopy detecting the triad of endometrial micropolyposis, stromal edema, and focal/diffuse hyperemia for the presence of histopathologic CE were 98.4% and 94.5%, respectively [32]. Based on these studies, the detection of endometrial micropolyposis using fluid hysteroscopy may represent a potentially useful diagnostic tool for CE in infertile women.

### 4.5. Diagnostic Significance and Interpretation of Plasma Cells

Our systematic review showed high heterogeneity in the selection criteria and plasma cell threshold used among the several studies. This heterogeneity explains the discrepancy in the reported prevalence of CE across studies. Several studies have demonstrated that the hysteroscopic evidence of mucosal edema, abundant micro-polypoid tissue, and hyperemia is highly predictive of CE with an accuracy rate of 92.7% [33,34,35]. However, although hysteroscopy has been proposed as an alternate method for the diagnosis of CE, endometrial hysteroscopic biopsy with histopathologic evidence of plasma cell infiltrate remains the gold standard and the most widely adopted method for the diagnosis of CE [33,34,35]. Plasma cells were first documented in endometria of patients with pelvic inflammatory disease and postpartum infection by Hitschman and Adler in 1907 [36]. In recent years, the increased use of Syndecan-1 (CD138), an immunohistochemical marker expressed on the plasma cell surface, has improved the pathologist’s ability to detect plasma cells in endometrial bioptic samples [15]. However, several studies highlighted that low levels of plasma cell infiltrate could also be observed in the endometrial tissue of healthy women [15,37,38]. Therefore, a more accurate and standardized threshold is needed for the diagnosis of CE and for predicting pregnancy outcomes. To date, there is no consensus on the more appropriate histological plasma cell threshold for the diagnosis of CE; the most frequently adopted thresholds include 1 plasma cell per 10 high-power fields (HPFs), 5 plasma cells dispersed over 10 HPFs, and 10 plasma cells per 10 HPFs, which were reported in the literature for the diagnosis of CE [1,2,13,15,21,22,23,24,25,39]. In addition, other types of cut-offs have been associated with infertility, RM, and RIF, such as ≥1 plasma cell/section [11,21,26,40,41,42], ≥5 plasma cells/20 HPFs, 1–5/HPF or discrete clusters, ≥1/HPF2, ESPDI (endometrial stromal plasmacyte density index, as the sum of the stromal CD138+ cell counts divided by the number of the high-power fields evaluated) ≥0.2537, and plasma cell density (plasma cell count per unit area, calculated from the entire area of the specimen, consisting of all fields whether complete or not) >5.15 cells/10 mm^2^ [43]. Moreover, most of these proposed thresholds have been selected arbitrarily (often not based on reference ranges derived from normal fertile populations) and tested in highly heterogeneous populations, in this way resulting in the overestimation of the CE prevalence rates. 

No consensus has been reached on the diagnostic value and the feasibility of additional morphological features classically associated with CE but not easily quantifiable (superficial stroma edema, stromal inflammatory infiltrate, increased stromal density, focal stromal hemorrhage, and spindling of stroma, most notably in the upper half of the mucosa) [23]. Another notable issue is represented by the histological sample collection timing that, in different stages of the cycle, could be a confounding variable. Plasma cells are often studied in the preovulatory phase since most intrauterine procedures in infertile women are performed in the preovulatory setting. According to several studies, the prevalence of CE is higher when the endometrial biopsy is obtained in the proliferative phase than in the postovulatory phase [44,45,46]. B lymphocytes, which differentiate into plasma cells, are mainly located in the basal layer of the endometrium; since the functional layer of the endometrium is thinner during the proliferative phase, there is a higher probability of obtaining higher endometrial stromal plasma cell density if endometrial biopsies are taken during the proliferative phase. 

It is well known that variances in endometrial sampling dates potentially cause biases in the diagnosis of histopathologic CE. According to a recent study by Ryan E et al., the likelihood of observing plasma cells in the early follicular phase (cycle days 5–8) is higher than in the late follicular phase (cycle days 9–14) and in the luteal phase [47]. 

The limitations of the study are the lack of a healthy control group; the heterogeneity of the study population that differed in terms of age, infertility history, and biopsy indication; and the retrospective nature of the study [48]. On this wave, potential confounding factors for diagnosing CE, such as the timing of the endometrial biopsy, should be considered when establishing a universally acceptable set of diagnostic guidelines for CE. Future, more reproducible studies and meta-analyses are still needed to identify the optimal biopsy timing to standardize the clinic-pathological diagnosis.

### 4.6. Recommendations for the Diagnosis of CE

Based on the currently published data and on our meta-analysis results, the following recommendations for the diagnosis of CE are suggested:-A threshold of 5 or more plasma cells in 10 HPFs seems more appropriate to diagnose CE.-The diagnosis of CE in patients with low or borderline plasma cell values (<5/10 HPF) should be integrated with clinical history and endometritis-associated hysteroscopic findings: the strawberry aspect of the endometrial lining, hyperemia, mycropolips, and mucosal edema.-In order to reduce possible variance in pathological results due to cyclic changes in the cell count, it is useful to know at the time of the histological diagnosis the precise menstrual phase.

### 4.7. Post-Treatment Diagnosis of Recurrent or Persistent CE

Data on the optimal therapeutical strategy for chronic endometritis are scarce. Broad-spectrum antibiotics are considered the conventional treatment, with various regimens and repeated cycles applied across studies. However, their indiscriminate administration can potentially cause the disruption of healthy uterine bacterial microbiomes and the development of antibiotic resistance, representing a major public health concern and additional expenses for any healthcare system. A recent retrospective case-control study evaluated the effectiveness of personalized, antibiogram-guided antibiotic treatment for CE in order to reach a good cumulative cure rate (81.3% after 3 antibiotic cycles), at the same time avoiding the antibiotic resistance phenomenon [49]. The overall cure rates range between 64% and 100%, but the definition of a cure differs between the studies [50]. In particular, women have been considered cured with plasma cell–negative histology/immunohistochemistry or a normalization of the hysteroscopy [23,49]. Similarly, in a study by Yang et al., those patients in whom control hysteroscopy showed the disappearance of CE “signs” had greater IVF success compared to women in whom immunohistochemistry demonstrated CE cure (i.e., no residual plasma cells), highlighting the hysteroscopic role in assessing the clinical effectiveness of antibiotic therapy [14].

Other studies have investigated alternative therapeutical options, such as anti-inflammatory drugs, progestogens, and probiotics, but the current evidence is insufficient to consider and apply them in daily clinical practice. We retain that the understanding of the correlation between CE therapy and female fertility/obstetric outcome appears to be the most challenging matter. Future multicentric, randomized studies should aim to investigate the effect of antibiotic therapeutical regimens not only on endometrial histology and the disappearance of plasma cells as a surrogate outcome parameter but also on reproductive success.

### 4.8. The Role of Microbial Culture 

Bacterial culture of the endometrium represents a useful diagnostic tool for the diagnosis of chronic endometritis [36]. This technique allows the identification of specific endometrial pathogens, thus providing useful information for the prescription of targeted therapies. Its introduction in clinical practice has significantly improved the reproductive outcomes in women with recurrent miscarriages and repeat implantation failure [36]. The main limitations related to endometrial culture include (i) risk of contamination from vaginal bacteria; (ii) limited culturability of selected microorganisms; (iii) delays in culturing bacteria; and (iv) long turnaround time [51]. Recently, RT-PCR has been demonstrated as a more efficient and less time-consuming alternative to bacterial culture for the identification of microorganisms responsible for chronic endometritis [51]. A recent prospective cohort study by Moreno et al. conducted in endometrial samples taken from patients with suspected chronic endometritis demonstrated the limitations of chronic endometritis diagnosis with classic culture techniques [36]. The authors demonstrated that RT-PCR testing outperformed other traditional methods in the diagnosis of chronic endometritis. RT-PCR was also capable of detecting intrauterine microbiome when histology is negative [36]. However, additional studies on larger cohorts are needed before introducing the molecular diagnosis of CE in daily diagnostic practice.

### 4.9. Imaging Techniques

Ultrasound examination in patients affected by chronic endometritis is usually associated with a thin endometrium showing hyperechoic areas related to calcification or fibrosis [52]. Hysterosalpingography is a useful technique in the evaluation of tubal status in infertile women since it can easily detect calcifications and irregularity of the endometrial cavity, fibrosis, and scarring [52]. However, it cannot be performed if there is a clinical suspicion of acute infection. 

### 4.10. Strengths and Limitations

To the best of our knowledge, our study represents the first meta-analysis evaluating the diagnostic plasma cells threshold for the diagnosis and management of patients affected by CE. Originality, search methodology, and the low risk of bias of the included studies are the main strengths of our paper. On the other hand, the small number of patients and the limited number of studies included in specific analyses, heterogeneity in patients’ clinical characteristics and biopsy timing, and the amount and random nature of endometrial tissue collection are the main limitations. 

## 5. Conclusions

In conclusion, our study suggests that the presence of plasma cells at low levels (<5/10 HPF) may be non-diagnostic of CE and may not predict pregnancy outcomes. Based on our findings, a threshold of 5 or more plasma cells in 10 HPFs might appear more appropriate to diagnose CE. For patients with low or borderline plasma cell values, the pathological diagnosis can be integrated with clinical history and endometritis-associated hysteroscopic findings, as suggested by the recommendations of the Working Group for Standardization of Chronic Endometritis Diagnosis: strawberry aspect of the endometrial lining, hyperemia, micro-polyps, and mucosal edema. Moreover, in order to reduce possible variance in pathological results due to cyclic changes in the cell count, it is useful to know at the time of the histological diagnosis the precise chronologic date (preferably 7 days after the LH surge). However, our results are limited by the heterogeneity found in the included studies, as well as the low number of studies for each plasma cell threshold. Future studies on larger series of infertile women are needed to determine the most reliable histopathologic features, microbiological changes, and clinical parameters for the correct integrated diagnosis of CE.

## Figures and Tables

**Figure 1 biomedicines-11-01714-f001:**
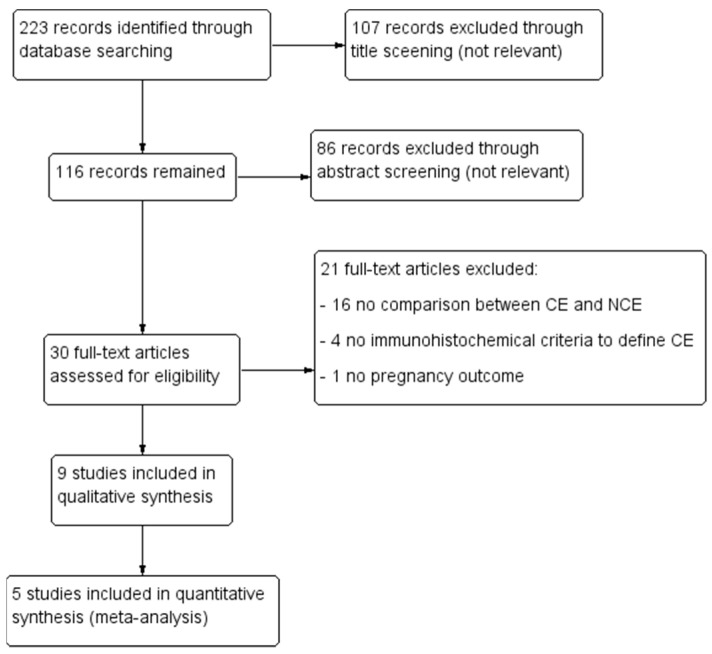
Literature search strategy.

**Figure 2 biomedicines-11-01714-f002:**
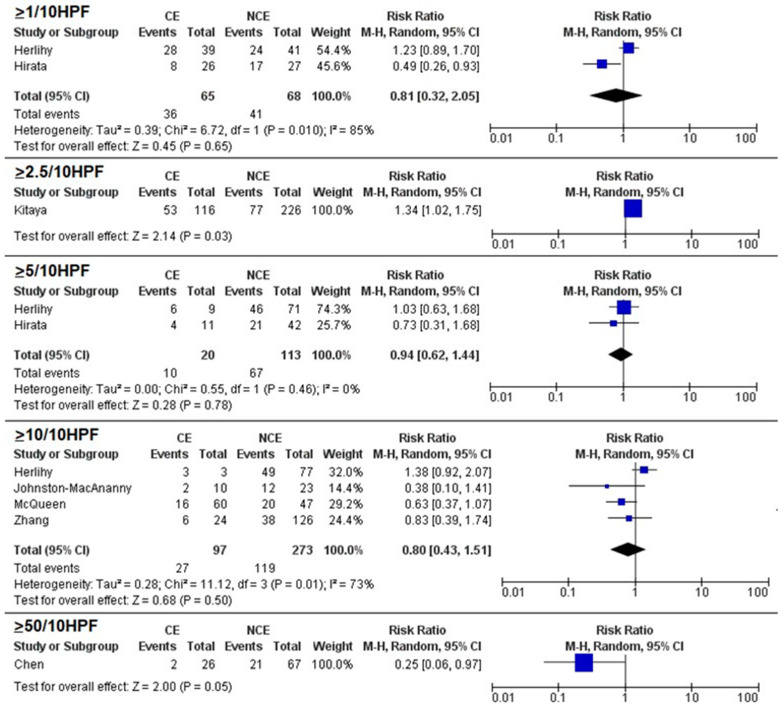
Forest plots assessing the difference in pregnancy rate between women with chronic endometritis (CE) and women with no chronic endometritis (NCE) according to different plasma cell thresholds. Blue squares are indicative of Risk Ratio for study; black diamonds are indicative of Risk Ratio for all the studies. ([1]: Chen, 2016; [2]: Johnston-MacAnanny, 2010; [15]: Herllihy, 2022; [24]: Zhang, 2019; [25]: Hirata, 2021; [26]: McQueen, 2021).

**Figure 3 biomedicines-11-01714-f003:**
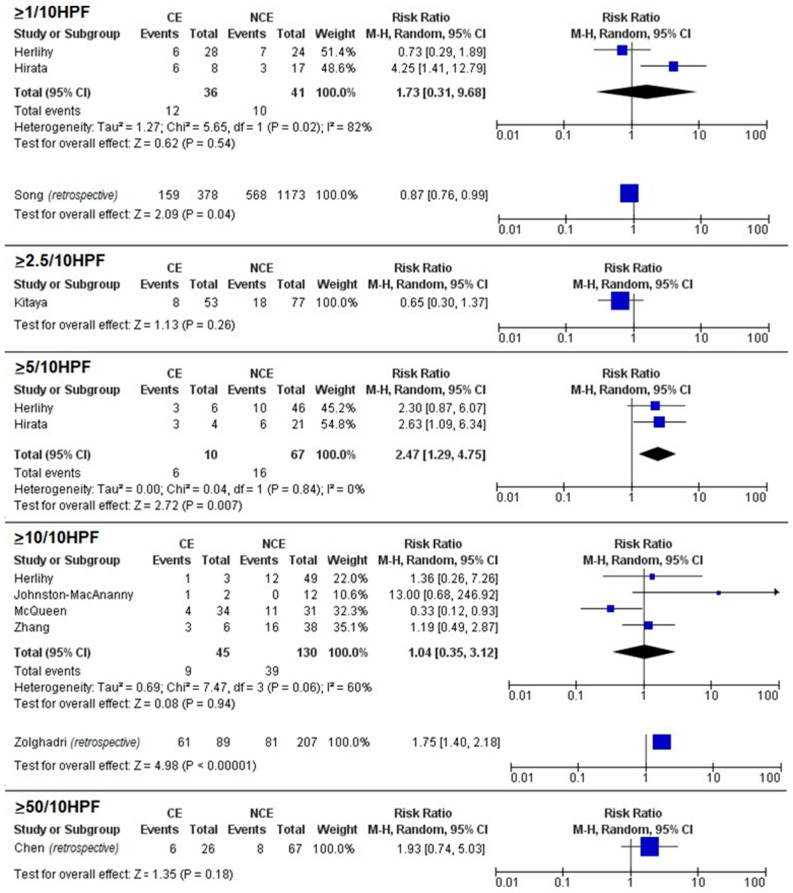
Forest plots assessing the difference in miscarriage rate between women with chronic endometritis (CE) and women with no chronic endometritis (NCE) according to different plasma cell thresholds. Blue squares are indicative of Risk Ratio for study; black diamonds are indicative of Risk Ratio for all the studies. ([1]: Chen, 2016; [2]: Johnston-MacAnanny, 2010; [15]: Herllihy, 2022; [24]: Zhang, 2019; [25]: Hirata, 2021; [26]: McQueen, 2021).

**Figure 4 biomedicines-11-01714-f004:**
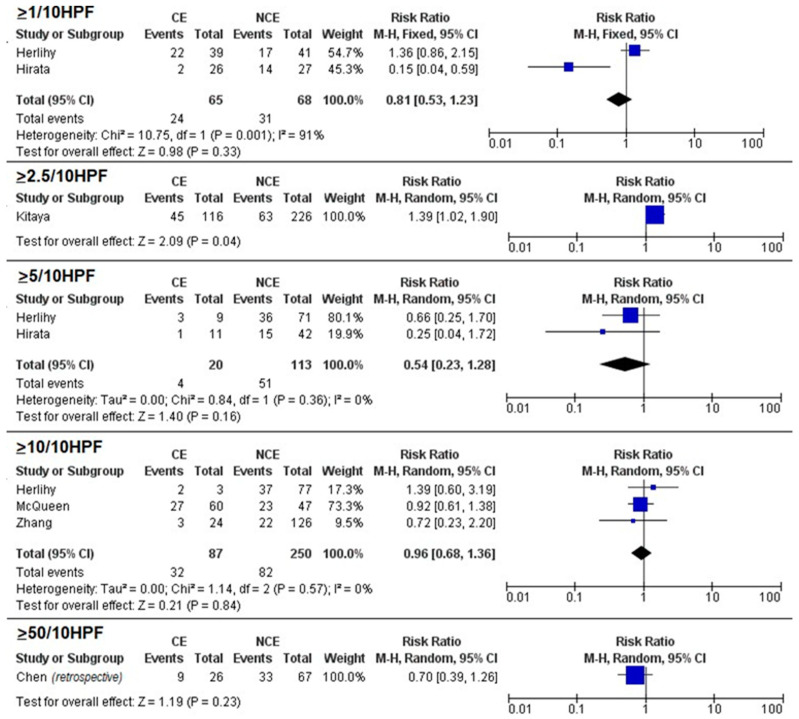
Forest plots assessing the difference in live birth rate between women with chronic endometritis (CE) and women with no chronic endometritis (NCE) according to different plasma cell thresholds. Blue squares are indicative of Risk Ratio for study; black diamonds are indicative of Risk Ratio for all the studies. ([1]: Chen, 2016; [2]: Johnston-MacAnanny, 2010; [15]: Herllihy, 2022; [24]: Zhang, 2019; [25]: Hirata, 2021; [26]: McQueen, 2021).

**Table 1 biomedicines-11-01714-t001:** Characteristics of the included studies.

Study	Country	Selection Criteria	Sample Size	Chronic Endometritis Frequency	Cd138 Threshold	Outcome Assessed
Johnston MacAnanny et al., 2009 [2]	USA	IVF-ET women with RIF	33	30.3%	1/HPF (10/10 HPF)	Pregnancy and miscarriage
Zolghadri et al., 2010 [21]	Iran	Women with unexplained recurrent spontaneous abortion	142	42.9%	1/HPF (10/10 HPF)	Miscarriage retrospectively assessed
McQueen et al., 2015 [22]	USA	Women with two or more pregnancy losses	107	56%	1/HPF (10/10 HPF)	Spontaneous pregnancy, live birth, and miscarriage
Chen et al., 2016 [1]	China	Assisted conception treatments	93	27.96%	5/HPF (50/10 HPF)	Pregnancy, miscarriage, and live birth retrospectively assessed
Kitaya et al., 2017 [23]	Japan	Women with a history of RIF	438	33.7%	0.25/HPF (2.5/10 HPF)	Pregnancy, live birth, and miscarriage
Song et al., 2018 [13]	China	Premenopausal women undergoing hysteroscopic biopsy	1551	24.4%	1/10 HPF	Miscarriage retrospectively assessed
Zhang et al., 2019 [24]	China	Women with a history of RIF	298	36.58%	1/HPF (10/10 HPF)	Pregnancy, live birth, and miscarriage
Hirata et al., 2021 [25]	Japan	Women who underwent single frozen-thawed blastocyst transfer with a hormone replacement cycle after histological examination	53	20.8% to 49.1%	1/10 HPF 2/10 HPF 3/10 HPF 5/10 HPF	Pregnancy, live birth, and miscarriage
Herlihy et al., 2022 [15]	USA	Infertile women undergoing IVF	80	4% to 49%	1/10 HPF 5/10 HPF 10/10 HPF	Pregnancy, live birth, and miscarriage

## Data Availability

Additional data are available from the corresponding author upon reasonable request.

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
