# Peer review of "The Role of Plasma Cells as a Marker of Chronic Endometritis: A Systematic Review and Meta-Analysis"

_biomedicines, 2023, doi:10.3390/biomedicines11061714_

Round 1

Reviewer 1 Report

Biomedicines-2296995-peer-review-v1

Angela Santoro et al.

The role of plasma cells as a marker of chronic endometritis: a systematic review and meta-analysis

The study designs seem to be safe and sound.

Critical comments

Line 62: Another histopathologic characteristics of CE include superficial edema, increased stromal density, and unsynchronized differentiation between endometrial epithelium and stroma.

Line 66: Chlamydia trachomatis (chlamydia) and Neisseria gonorrhoeae (gonorrhea) are not major pathogens for CE. Common bacteria (Streptococcus. Enterococcus, Escherichia, and Staphylococcus) are much more prevalent.

Discussion: The studies that obtained endometrial biopsy in the proliferative phase report higher endometrial stromal plasma cell density than the studies that did in the secretory phase. Please discuss that the variances in endometrial sampling date potentially causes biases in the diagnosis of histopathologic CE and precautions are required in this kind of meta-analysis.

Author Response

We thank the reviewer for all comments and suggestions.

All changes in the manuscript have been highlighted in yellow.

Q1: Line 62: Another histopathologic characteristics of CE include superficial edema, increased stromal density, and unsynchronized differentiation between endometrial epithelium and stroma.

A1: As suggested, we have also added the other histopathological features of CE

Q2: Line 66: Chlamydia trachomatis (chlamydia) and Neisseria gonorrhoeae (gonorrhea) are not major pathogens for CE. Common bacteria (Streptococcus. Enterococcus, Escherichia, and Staphylococcus) are much more prevalent.

A2: We have modified, according to your suggestion

Q3: Discussion: The studies that obtained endometrial biopsy in the proliferative phase report higher endometrial stromal plasma cell density than the studies that did in the secretory phase. Please discuss that the variances in endometrial sampling date potentially causes biases in the diagnosis of histopathologic CE and precautions are required in this kind of meta-analysis.

A3: We have commented about the endometrial biopsy timing as important potential diagnostic bias.

Reviewer 2 Report

This is a useful and well-conducted meta-analysis with clinically far-reaching results. The topic is relevant, the methods are appropriately selected and described.

 1. I recommend a better presentation of the results and conclusions regarding:

 1.1. Phase of the menstrual cycle, in particular the threshold in the proliferative phase; plasma cells are often additionally studied in the preovulatory phase, since most intrauterine procedures in infertile women are performed preovulatory. In contrast, most studies limit the value of plasma cells to diagnosis in the postovulatory phase.

Please, see and discuss the related publication:

a) Ryan et al. The menstrual cycle phase impacts the detection of plasma cells and the diagnosis of chronic endometritis in endometrial biopsy specimens. Fertil Steril. 2022. doi: 10.1016/j.fertnstert.2022.07.011. PMID: 36182264.

And the comment on it:

b) Sokalska A. Timing of endometrial biopsy: Are we one step closer to the definition of chronic endometritis? Fertil Steril. 2022. doi: 10.1016/j.fertnstert.2022.08.002. PMID: 36109251.

1.2. Post-treatment diagnosis of recurrent or persistent CE.

2. Restricting the included studies to those published before February 2022 introduces a bias as more studies have since been published on this topic (and using a different cutoff 5/HPF), e.g.

a) Li et al. Front Cell Dev Biol. 2023 Feb 13;11:1088586 doi:10.3389/fcell.2023.1088586 PMID:36861040 or

b) Li et al Taiwan J Obstet Gynecol 2022 doi:10.1016/j.tjog.2021.01.034 PMID: 36428002

3. Since the paper has the potential to serve as a reference for everyday practice, the paragraph (4.5.) on the plasma cell threshold should not be a simple discussion of the literature but should contain clearly formulated recommendations.

4. a) Line 375: The meaning of the sentence is not clear.

b) The formatting of the tables should be improved.

c) The format of the references is a bit chaotic and should be improved according to the standards of the journal

Author Response

We thank the reviewer for all comments and suggestions.

All changes in the manuscript have been highlighted in yellow.

Q1. I recommend a better presentation of the results and conclusions regarding:

Q1.1. Phase of the menstrual cycle, in particular the threshold in the proliferative phase; plasma cells are often additionally studied in the preovulatory phase, since most intrauterine procedures in infertile women are performed preovulatory. In contrast, most studies limit the value of plasma cells to diagnosis in the postovulatory phase.

Please, see and discuss the related publication:

  1. a) Ryan et al. The menstrual cycle phase impacts the detection of plasma cells and the diagnosis of chronic endometritis in endometrial biopsy specimens. Fertil Steril. 2022. doi: 10.1016/j.fertnstert.2022.07.011. PMID: 36182264.

And the comment on it:

  1. b) Sokalska A. Timing of endometrial biopsy: Are we one step closer to the definition of chronic endometritis? Fertil Steril. 2022. doi: 10.1016/j.fertnstert.2022.08.002. PMID: 36109251.

A1.1. We have commented this publications about the endometrial biopsy timing as important potential diagnostic bias

Q1.2. Post-treatment diagnosis of recurrent or persistent CE.

A1.2. We have commented about this topic in a new added separate paragraph

  1. Restricting the included studies to those published before February 2022 introduces a bias as more studies have since been published on this topic (and using a different cutoff 5/HPF), e.g.
  2. a) Li et al. Front Cell Dev Biol. 2023 Feb 13;11:1088586 doi:10.3389/fcell.2023.1088586 PMID:36861040 or
  3. b) Li et al Taiwan J Obstet Gynecol 2022 doi:10.1016/j.tjog.2021.01.034 PMID: 36428002

A2. According to the reviewer suggestion we have included in the Discussion section comments regarding these more recent publications. We retain that future meta-analyses should selectively collect RIF patients, with particular attention to treated women, in order to define the impact of cured CE on reproductive success.

Q3. Since the paper has the potential to serve as a reference for everyday practice, the paragraph (4.5.) on the plasma cell threshold should not be a simple discussion of the literature but should contain clearly formulated recommendations.

A3. As suggested, we included an additional paragraph (4.6 Recommendations for the diagnosis of CE) to better elucidate the pathological and clinical features consistent with the diagnosis of CE.

Q4.

  1. a) Line 375: The meaning of the sentence is not clear: Up to date, several diagnostic thresholds such as 1 plasma cell per 10 high-power fields (HPFs), 5 plasma cells dispersed over 10 HPFs, and 10 plasma cells per 10 HPFs, have been reported in the literature for the diagnosis of CE

A4a: The sentence has been rewritten in order to better elucidate the different histological thresholds adopted fotr the diagnosis of CE.

  1. b) The formatting of the tables should be improved.

A4b; Table has been re-formatted as suggested.

  1. c) The format of the references is a bit chaotic and should be improved according to the standards of the journal

A4c: all references have been re-formatted according to standards of the journal.